# NON-MARKOVIAN PREDICTIVE CODING FOR PLANNING IN LATENT SPACE

## ABSTRACT

High-dimensional observations are a major challenge in the application of model-based reinforcement learning (MBRL) to real-world environments. In order to handle high-dimensional sensory inputs, existing MBRL approaches use representation learning to map high-dimensional observations into a lower-dimensional latent space that is more amenable to dynamics estimation and planning. Crucially, the task-relevance and predictability of the learned representations play critical roles in the success of planning in latent space. In this work, we present Non-Markovian Predictive Coding (NMPC), an information-theoretic approach for planning from high-dimensional observations with two key properties: 1) it formulates a mutual information objective that prioritizes the encoding of task-relevant components of the environment; and 2) it employs a recurrent neural network capable of modeling non-Markovian latent dynamics. To demonstrate NMPC's ability to prioritize task-relevant information, we evaluate our new model on a challenging modification of standard DMControl tasks where the DMControl background is replaced with natural videos, containing complex but irrelevant information to the planning task. Our experiments show that NMPC is superior to existing methods in the challenging complex-background setting while remaining competitive with current state-of-the-art MBRL models in the standard setting.

## 1 INTRODUCTION

Learning to control from high dimensional observations has been made possible due to the advancements in reinforcement learning (RL) and deep learning. These advancements have enabled notable successes such as solving video games (Mnih et al., 2015; Lample & Chaplot, 2017) and continuous control problems (Lillicrap et al., 2016) from pixels. However, it is well known that performing RL directly in the high-dimensional observation space is sample-inefficient and may require a large amount of training data (Lake et al., 2017). This is a critical problem, especially for real-world applications. Recent model-based RL works (Kaiser et al., 2020; Ha & Schmidhuber, 2018; Hafner et al., 2019; Zhang et al., 2019; Hafner et al., 2020) proposed to tackle this problem by learning a world model in the latent space, and then applying RL algorithms in the latent world model.

The existing MBRL methods that learn a latent world model typically do so via reconstruction-based objectives, which are likely to encode task-irrelevant information, such as of the background. In this work, we take inspiration from the success of contrastive learning and propose Non-Markovian Predictive Coding (NMPC), a novel information-theoretic approach for planning from pixels. In contrast to reconstruction, NMPC formulates a mutual information (MI) objective to learn the latent space for control. This objective circumvents the need to reconstruct and prioritizes the encoding of task-relevant components of the environment, thus make NMPC more robust when dealing with complicated observations. Our primary contributions are as follows:

- We propose Non-Markovian Predictive Coding (NMPC), a novel information-theoretic approach to learn latent world models for planning from high-dimensional observations and theoretically analyze its ability to prioritize the encoding of task-relevant information.

- We show experimentally that NMPC outperforms the state-of-the-art model when dealing with complex environments dominated by task-irrelevant information, while remaining competitive on standard DeepMind control (DMControl) tasks. Additionally, we conduct detailed ablation analyses to study the empirical importance of the components in NMPC.

## 2 BACKGROUND

The motivation and design of our model are largely based on two previous works (Shu et al., 2020; Hafner et al., 2020). In this section, we briefly go over the key concepts in each work. Shu et al. (2020) proposed PC3, an information-theoretic approach that uses contrastive predictive coding (CPC) to learn a latent space amenable to locally-linear control. Specifically, they present the theory of predictive suboptimality to motivate a CPC objective between the latent states of two consecutive time steps, instead of CPC between the frame and its corresponding state. Moreover, they use the latent dynamics $F$ as the variational device in the lower bound

$$\ell_{\text{cpc}}(E, F) = \mathbb{E}\frac{1}{K} \sum_i \ln \frac{F\left(E\left(o_{t+1}^{(i)}\right) \mid E\left(o_t^{(i)}\right), a_t^{(i)}\right)}{\frac{1}{K}\sum_j F\left(E\left(o_{t+1}^{(i)}\right) \mid E\left(o_t^{(j)}\right), a_t^{(j)}\right)} \tag{1}$$

This particular design of the critic has two benefits, where it circumvents the instantiation of an auxiliary critic, and also takes advantage of the property that an optimal critic is the true latent dynamics. However, the author also shows that this objective does not ensure the learning of a latent dynamics $F$ that is consistent with the true latent dynamics, therefore introduces the consistency loss to ensure the latent dynamics model $F$ indeed approximates the true latent dynamics.

$$\ell_{\text{cons}}(E, F) = \mathbb{E}_{p(o_{t+1}, o_t, a_t)} \ln F\left(E\left(o_{t+1}\right) \mid E\left(o_t\right), a_t\right) \tag{2}$$

Since PC3 only tackles the problem from an optimal control perspective, it is not readily applicable to RL problems. Indeed, PC3 requires a depiction of the goal image in order to perform control, and also the ability to teleport to random locations of the state space to collect data, which are impractical in many problems. On the other hand, Dreamer (Hafner et al., 2020) achieves state-of-the-art performance on many RL tasks, but learns the latent space using a reconstruction-based objective. While the authors included a demonstration of a contrastive approach that yielded inferior performance to their reconstruction-based approach, their contrastive model applied CPC between the frame and its corresponding state, as opposed to between latent states across time steps.

In this paper, we present Non-Markovian Predictive Coding (NMPC), a novel latent world model that leverages the concepts in PC3 and apply them to Dream paradigm and RL setting. Motivated by PC3, we formulate a mutual information objective between historical and future latent states to learn the latent space. We additionally take advantage of the recurrent model in Dreamer to model more complicated dynamics than what was considered in PC3. The use of recurrent dynamics also allows us to extend Eqs. (1) and (2) to the non-Markovian setting, which was not considered in PC3.

## 3 NON-MARKOVIAN PREDICTIVE CODING FOR PLANNING FROM PIXELS

To plan in an unknown environment, we need to model the environment dynamics from experience. We do so by iteratively collecting new data and using those data to train the world model. In this section, we focus on presenting the proposed latent world model, its components and objective functions, and provide practical considerations when implementing the method.

### 3.1 NON-MARKOVIAN PREDICTIVE CODING

We aim to learn a latent dynamics model for planning. To do that, we define an encoder $E$ to embed high-dimensional observations into a latent space, a latent dynamics $F$ to model the world in this space, and a reward function, as follows

$$\begin{aligned}
\text{Encoder:} \quad & E(o_t) = s_t \\
\text{Latent dynamics:} \quad & F(s_t \mid s_{<t}, a_{<t}) = p(s_t \mid s_{<t}, a_{<t}) \\
\text{Reward function:} \quad & R(r_t \mid s_t) = p(r_t \mid s_t)
\end{aligned} \tag{3}$$

in which $t$ is the discrete time step, $\{o_t, a_t, r_t\}_{t=1}^T$ are data sequences with image observations $o_t$, continuous action vectors $a_t$, scalar rewards $r_t$, and $s_t$ denotes the latent state at time t. To handle potentially non-Markovian environment dynamics, we model the transition dynamics using a recurrent

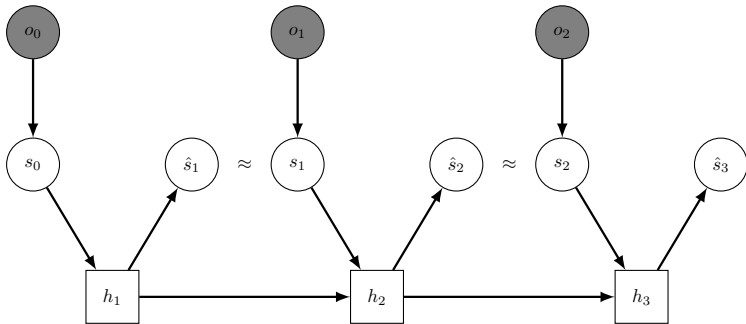

Figure 1: The graphical model of NMPC, in which we employ a recurrent neural network to model the dynamics. We omit the action and reward for simplicity. The approximation of $\hat{s}_t$ with $s_t$ is done by using a combination of contrastive predictive coding and maximum likelihood.

neural network with a deterministic state $h_t = \text{RNN}(h_{t-1}, s_{t-1}, a_{t-1})$, which summarizes information about the past, followed by the stochastic state model $p(s_t|s_{<t}, a_{<t}) = p(s_t|h_t)$. In practice, we use a deterministic encoder, and Gaussian distribution for dynamics and reward functions. The graphical model is presented in Figure 1. We now introduce the three key components of our model: *recurrent contrastive predictive coding*, *non-markovian consistency*, and *reward prediction*.

**Recurrent contrastive predictive coding**  Instead of performing pixel prediction to learn $E$ and $F$ as in Hafner et al. (2019; 2020), we take a different approach, where we maximize the mutual information (MI) between the past latent codes and actions against the future latent code $I(E(O_t); E(O_{<t}), A_{<t})$. This objective prioritizes the encoding of predictable components from the environment, which potentially helps avoid encoding nuance information when dealing with complex image observations. A similar MI objective was previously used in Shu et al. (2020). However, Shu et al. (2020) only considered the Markovian setting and thus focused on the mutual information over consecutive time steps. In contrast, we consider a non-Markovian setting and wish to maximize the mutual information between the latent code at any time step $t$ and the entire historical trajectory latent codes and actions prior to $t$. Given a trajectory over $T$ steps, our non-Markovian MI objective thus sums over every possible choice of $t \in \{2, \ldots, T\}$,

$$\sum_{t=2}^{T} I(E(O_t); E(O_{<t}), A_{<t}). \tag{4}$$

To estimate this quantity, we employ contrastive predictive coding (CPC) proposed by Oord et al. (2018). We perform CPC by introducing a critic function $f$ to construct the lower bound at a particular time step $t$,

$$I(E(O_t); E(O_{<t}), A_{<t}) \geq \mathbb{E}\frac{1}{K}\sum_i \ln \frac{\exp f(E(o_t^{(i)}), E(o_{<t}^{(i)}), a_{<t}^{(i)})}{\frac{1}{K}\sum_j \exp f(E(o_t^{(i)}), E(o_{<t}^{(j)}), a_{<t}^{(j)})} =: \ell_{\text{r-cpc}}^{(t)}, \tag{5}$$

where the expectation is over $K$ i.i.d. samples of $(o_t, o_{<t}, a_{<t})$. Note that Eq. (5) uses past $(E(o_{<t}^{(j)}), a_{<t}^{(j)})$ from unrelated trajectories as an efficient source of negative samples for the contrastive prediction of the future latent code $E(o_t^{(i)})$. Following Shu et al. (2020), we choose to tie $f$ to our recurrent latent dynamics model $F$,

$$\exp f(s_t, s_{<t}, a_{<t}) = F(s_t|s_{<t}, a_{<t}). \tag{6}$$

There are two favorable properties of this particular design. First, it is parameter-efficient since we can circumvent the instantiation of a separate critic $f$. Moreover, it takes advantage of the fact that an optimal critic is the true latent dynamics induced by the encoder $E$ (Poole et al., 2019; Ma & Collins, 2018). We denote our overall CPC objective as $\ell_{\text{r-cpc}}(E, F) = \sum_{t=2}^{T} \ell_{\text{r-cpc}}^{(t)}(E, F)$.

**Non-Markovian consistency**  Although the true dynamics is an optimal critic for the CPC bound, maximizing this objective only does not ensure the learning of a latent dynamics model $F$ that

is consistent with the true latent dynamics, due to the non-uniqueness of the optimal critic (Shu et al., 2020). Since an accurate dynamics is crucial for planning in the latent space, we additionally introduce a consistency objective, which encourages the latent dynamics model to maintain a good prediction of the future latent code given the past latent codes and actions. Similar to the recurrent CPC objective, we optimize for consistency at every time step in the trajectory,

$$\ell_{\text{cons}}(E, F) = \sum_{t=2}^{T} \mathbb{E}_{p(o_t, o_{<t}, a_{<t})} \ln F(E(o_t) | E(o_{<t}), a_{<t}). \tag{7}$$

**Reward prediction** Finally, we train the reward function by maximizing the likelihood of the true reward value conditioned on the latent state $s_t = E(o_t)$,

$$\ell_{\text{reward}}(E, R) = \sum_{t=1}^{T} \mathbb{E}_{p(o_t)} \ln R(r_t | E(o_t)). \tag{8}$$

## 3.2 THEORETICAL ANALYSIS OF NMPC AND TASK-RELEVANCE

In contrast to a reconstruction-based objective, which explicitly encourages the encoder to behave injectively on the space of observations, our choice of mutual information objective as specified in Eq. (5) may discard information from the observed scene. In this section, we wish to formally characterize the information discarded by our MI objective and argue that any information discarded by an optimal encoder under our MI objective is provably task-irrelevant.

**Lemma 1.** *Consider an optimal encoder and reward predictor pair $(E^*, R^*)$ where*

$$\arg\max_{E} I(E(O_t) ; E(O_{<t}), A_{<t}) = E^* \tag{9}$$

$$D_{KL}(p(r_t \mid o_t) \parallel R^*(r_t \mid E^*(o_t))) = 0. \tag{10}$$

*Let $\pi(a_t \mid E^*(o_{\leq t}), a_{<t})$ denote an $E^*$-restricted policy whose access to the observations $o_{<t}$ is restricted by $E^*$. Let $\pi_{\text{aux}}(a_t \mid E^*(o_{\leq t}), E'(o_{\leq t}), a_{<t})$ denote an $(E^*, E')$-restricted policy which has access to auxiliary information about $o_{<t}$ via some encoder $E'$. Let $\eta(\pi)$ denote the expected cumulative reward achieved by a policy $\pi$ over a finite horizon $T$. Then there exists no encoder $E'$ where the optimal $E^*$-restricted policy underperforms the optimal $(E^*, E')$-restricted policy,*

$$\nexists E' \text{ s.t. } \eta(\pi^*) < \eta(\pi_{\text{aux}}^*). \tag{11}$$

Intuitively, since $E^*$ optimizes our MI objective, any excess information contained in $E'$ about $o_t$ (not already accounted for by $E^*$) must be *temporally-unpredictive*—it is neither predictable from the past nor predictive of the future. The excess information conveyed by $E'$ is effectively nuisance information and thus cannot be exploited to improve the agent's performance. It is therefore permissible to dismiss the excess information in $E'$ as being task-irrelevant. We provide a proof formalizing this intuition in Appendix B.

It is worth noting that our MI objective does not actively penalize the encoding of nuisance information, nor does temporally-predictive information necessarily mean it will be task-relevant. However, if the representation space has limited capacity to encode information about $o_t$ (e.g., due to dimensionality reduction, a stochastic encoder, or an explicit information bottleneck regularizer), our MI objective will favor temporally-predictive information over temporally-unpredictive information— and in this sense, thus favor potentially task-relevant information over provably task-irrelevant information. This is in sharp contrast to a reconstruction objective, which makes no distinction between these two categories of information contained in the observation $o_t$.

## 3.3 PRACTICAL IMPLEMENTATION OF NMPC

**Avoiding map collapse** Naively optimizing $\ell_{\text{r-cpc}}$ and $\ell_{\text{cons}}$ can lead to a trivial solution, where the latent map collapses into a single point to increase the consistency objective arbitrarily. In the previous work, Shu et al. (2020) resolved this by adding Gaussian noise to the future encoding, which balances the latent space retraction encouraged by $\ell_{\text{cons}}$ with the latent space expansion encouraged by $\ell_{\text{r-cpc}}$. However, we found this trick insufficient, and instead introduce a CPC objective between

the current observation and its corresponding latent code, which we call the instantaneous CPC. Our instantaneous CPC objective defines a lower bound for $I(E(O_t); O_t)$ where we employ a Gaussian distribution $\mathcal{N}(E(O_t) \mid E(O_t), \sigma I)$ as our choice of variational critic for $f(E(O_t), O_t)$. Crucially, we keep the variance $\sigma$ fixed so that the instantaneous CPC objective encourages map expansion.

**Smoothing the dynamics model**  Since our encoder is deterministic, the dynamics always receives clean latent codes as inputs during training. However, in behavior learning, we roll out multiple steps towards the future from a stochastic dynamics. These roll outs are susceptible a cascading error problem, which hurts the value estimation and policy learning. To resolve this issue, we smooth the dynamics by adding Gaussian noise to the inputs of the recurrent dynamics during training. The noise-adding procedure is as follows: assume the dynamics outputs $\hat{s}_t \sim \mathcal{N}(\mu_t, \sigma_t^2)$ as the prediction at time step $t$, we then add $\epsilon_t \sim \mathcal{N}(0, \sigma_t^2)$ to $s_t = E(o_t)$ and feed it to the latent dynamics, and repeat for every time step $1 \leq t \leq T$. We call this dynamics-associated noise, which ensures that the latent dynamics can handle the amount of noise that it produces when rolling out.

The overall objective of our model is thus

$$\max_{E,F,R} \lambda_1 \ell_{\text{r-cpc}}(E, F) + \lambda_2 \ell_{\text{cons}}(E, F) + \lambda_3 \ell_{\text{i-cpc}}(E) + \lambda_4 \ell_{\text{reward}}(E, R). \tag{12}$$

## 4 BEHAVIOR LEARNING

Following Hafner et al. (2020), we use latent imagination to learn a parameterized policy for control. For self-containedness, in this section we give a summary of this approach. Given the latent state $s_t = E(o_t)$, we roll out multiple steps into the future using the learned dynamics $F$, estimate the return and perform backpropagation through the dynamics to maximize this return, which in turn improves the policy.

**Action and value models**  We define two components needed for behavior learning, the action model and the value model, which both operate on the latent space. The value model estimates the expected imagined return when following the action model from a particular state $s_\tau$, and the action model implements a policy, conditioned on $s_\tau$, that aims to maximize this return. With imagine horizon $H$, we have

$$\text{Action model:} \quad a_\tau \sim \pi(a_\tau | s_\tau) \qquad \text{Value model:} \quad v(s_\tau) \approx \mathbb{E}_{\pi(\cdot | s_\tau)} \sum_{\tau=t}^{t+H} \gamma^{\tau-t} r_\tau \tag{13}$$

**Value estimation**  To learn the action and value model, we need to estimate the state values of imagined trajectories $\{s_\tau, a_\tau, r_\tau\}_{\tau=t}^{t+H}$, where $s_\tau$ and $a_\tau$ are sampled according to the dynamics and the policy. In this work, we use value estimation presented in Sutton & Barto (2018),

$$V_N^k(s_\tau) \doteq E_{q_\theta, q_\phi} \left( \sum_{n=\tau}^{h-1} \gamma^{n-\tau} r_n + \gamma^{h-\tau} v_\psi(s_h) \right) \quad \text{with} \quad h = \min(\tau + k, t + H)$$

$$V_\lambda(s_\tau) \doteq (1 - \lambda) \sum_{n=1}^{H-1} \lambda^{n-1} V_N^n(s_\tau) + \lambda^{H-1} V_N^H(s_\tau). \tag{14}$$

which allows us to estimate the return beyond the imagine horizon. The value model is then optimized to regress this estimation, while the action model is trained to maximize the value estimation of all states $s_\tau$ along the imagined trajectories,

$$\max_\pi \mathbb{E} \left( \sum_{\tau=t}^{t+H} V_\lambda(s_\tau) \right) \qquad \min_v \mathbb{E} \left( \sum_{\tau=t}^{t+H} \frac{1}{2} \| v_\psi(s_\tau) - V_\lambda(s_\tau) \|^2 \right). \tag{15}$$

## 5 EXPERIMENTS

In this section, we empirically evaluate the proposed NMPC model in various settings. First, we design experiments to compare the relative performance of our model with the current best model-based method in several standard control tasks. Second, we evaluate its ability to handle a more

realistic but also more complicated scenario, in which we replace the background of the environment with a natural video. We discuss how our model is superior in the latter case, while remaining competitive in the standard setting. Finally, we conduct ablation studies to demonstrate the importance of the components of our model.

**Control tasks** For the standard setting, we test our model on 6 DeepMind Control (DMC) tasks (Tassa et al., 2018): Cartpole Swingup, Cheetah Run, Walker Run, Acrobot Swingup, Hopper Hop and Cup Catch. In the natural background setting, we replace the background of each data trajectory with a video taken from the kinetics dataset (Kay et al., 2017). We use a different set of videos for training and testing to also test the generalization of each method. In both settings, we use images of shape $64 \times 64 \times 3$, episodes with 1000 steps and action repeat 2 across tasks.

**Baseline methods** We compare NMPC with Dreamer (Hafner et al., 2020), the current state of the art model-based method for planning from pixels. Each method is evaluated by the environment return in 1000 steps. For Dreamer, we use the best set of hyperparameters as reported in their paper. We run each task with 3 different seeds.

## 5.1 COMPARISONS IN STANDARD SETTING

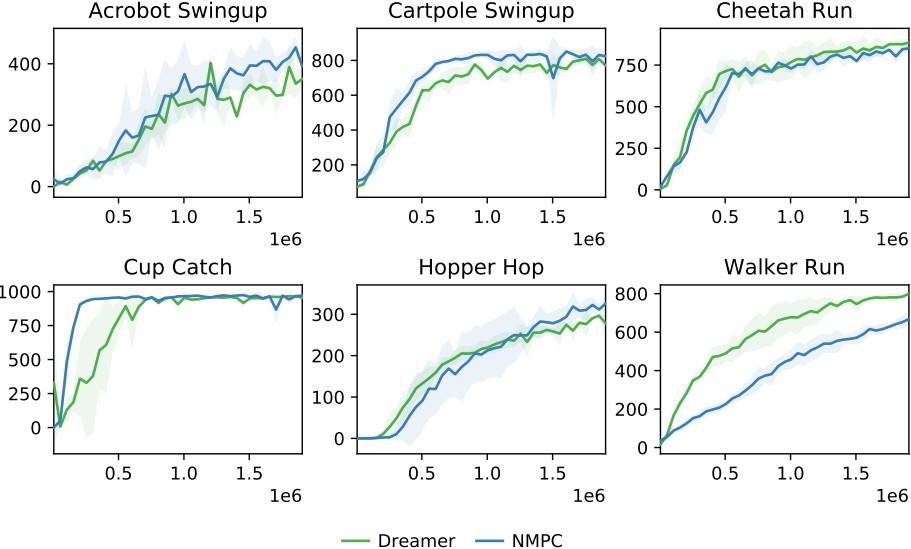

Figure 2: NMPC vs Dreamer in the standard DMControl tasks. The return is computed in 1000 environment steps. Each task is run with 3 seeds.

We demonstrate the performance of both methods on standard control tasks in Figure 2. NMPC is competitive in all the tasks, which is in contrast to what was previously observed in Dreamer (Hafner et al., 2020), where they showed the inferiority of a contrastive learning approach (which applies CPC between an image observation and its corresponding latent code) compared to their reconstruction-based approach. Our results thus show that the use of recurrent contrastive predictive coding is critical for learning a latent space that is suitable behavior learning.

## 5.2 COMPARISONS IN NATURAL BACKGROUND SETTING

In this setting, we evaluate the robustness of NMPC versus Dreamer in dealing with complicated, natural backgrounds. The performance of both models in the natural backgrounds setting is shown in Figure 4. Dreamer fails to achieve meaningful performance across all six tasks. NMPC performs significantly better on four of the six tasks, only failing to work on Acrobot Swingup and Hopper Hop. We note that Acrobot Swingup and Hopper HOp are the two most challenging tasks in the standard setting. Furthermore, the agents in these two tasks are also tiny compared to the complex backgrounds, thus making it difficult for the model to distill task-relevant information.

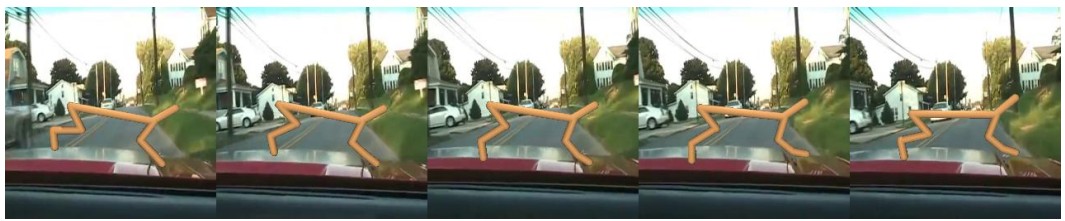

Figure 3: DeepMind Control tasks in the natural backgrounds setting. The images show a sample training trajectory in Cheetah Run task, where the background is taken from the kinetics dataset.

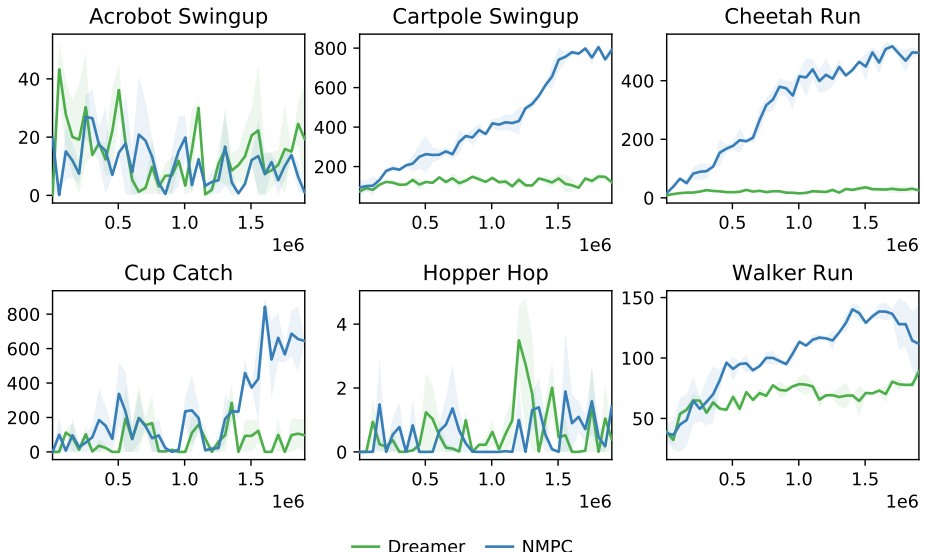

Figure 4: NMPC vs Dreamer in the natural backgrounds setting. The return is computed in 1000 evironment steps. Each task is run with 3 seeds.

## 5.3 ABLATION ANALYSIS

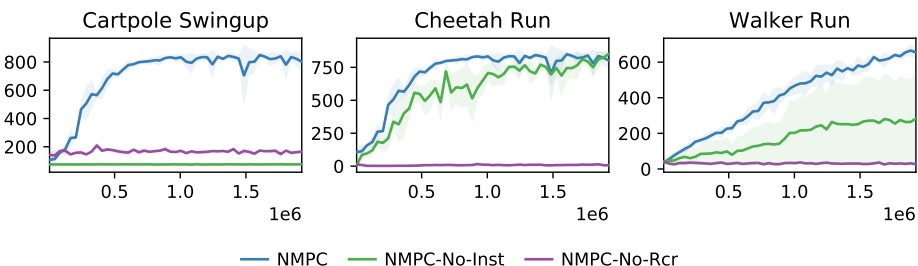

Figure 5: NMPC vs two different variants, where we omit the recurrent CPC in NMPC-No-Rcr and instantaneous CPC in NMPC-No-Inst. The return is computed in 1000 evironment steps. Each task is run with 3 seeds.

We conduct an ablation analysis to evaluate the importance of each CPC objective employed in NMPC. To do so, we compare the original model with two variants, where we omit the recurrent CPC or instantaneous CPC objective respectively. The relative performance of these models on three control tasks is shown in Figure 5. The original NMPC achieves the best performance, while the variant with only instantaneous CPC (denoted NPMC-No-Rcr) fails across all three tasks. NMPC without instantaneous CPC is unstable, since it faces the map collapsing problem, which leads to poor performance.

## 6 RELATED WORK

**Learning latent space for model-based RL via reconstruction**     Latent world models can be learned by jointly training the encoder and the latent dynamics model with observation reconstruction loss. Learning Controllable Embedding (LCE) approach, including E2C (Watter et al., 2015), RCE (Banijamali et al., 2018) and PCC (Levine et al., 2020), uses randomly collected data to pretrain a Markovian latent dynamic model that is specifically designed for locally-linear control, then run offline optimal control on top of the learned latent space. CARL (Cui et al., 2020) extends these works for Soft Actor-Critic (Haarnoja et al., 2018) and also proposes an online version, in which they iteratively learn the model and a parameterized policy. World Models (Ha & Schmidhuber, 2018) learn a recurrent latent dynamic model in a two-stage process to evolve their linear controllers in imagination. PlaNet (Hafner et al., 2019) jointly learns a recurrent state space model (RSSM) and plans in latent space using the cross entropy method, while Dreamer (Hafner et al., 2020) uses RSSM to iteratively learn the model and the policy by backpropagating through the dynamics. SO-LAR (Zhang et al., 2019) models the dynamics as time-varying linear-Gaussian with quadratic costs and controls using guided policy search. However, training world models with reconstruction loss has several drawbacks: it requires a decoder as an auxiliary network for predicting images, and by reconstructing every single pixel, those methods are potentially vulnerable to task-irrelevant information such as an irrelevant background.

**Learning latent space for model-based RL via constrastive learning**     An alternative framework for learning latent world models is contrastive learning. Contrastive learning is a self-supervised learning technique that aims to learn representations by contrasting positive samples against negative samples without having to reconstruct images (Oord et al., 2018; Chen et al., 2020a). Recently proposed contrastive learning methods have achieved significant successes in learning representations purely from unlabeled data, which include works by (Chen et al., 2020a;b; Bachman et al., 2019; Hénaff et al., 2019; He et al., 2020; Tian et al., 2019). Poole et al. (2019) has also established a close connection between contrastive learning and mutual information maximization. In the context of RL, recent works have proposed to use this framework to accelerate RL from pixels in two distinct directions: 1) cast contrastive learning as an auxiliary representation learning task, and use model-free RL methods on top of the learned latent space (Oord et al., 2018; Srinivas et al., 2020); and 2) use contrastive learning in conjunction with learning a latent dynamics for planning in the latent space (Shu et al., 2020; Ding et al., 2020; Hafner et al., 2020).

Our proposed method follows the latter direction, and can be seen as an extension of both PC3 (Shu et al., 2020) and Dreamer (Hafner et al., 2020). While having a similar contrastive objective to PC3, we design a non-Markovian version of it and also use a stronger controller, which enables our model to work on more complicated control tasks. We use the same controller as proposed in Dreamer; however, while their contrastive variant maximizes the MI between the observation its latent code, our loss maximizes the MI between the historical latent codes and actions versus the future latent code. Experimental results show that NMPC is competitive to Dreamer with reconstruction in standard DMControl tasks, which means we outperform their contrastive variant.

## 7 CONCLUSION

In this work, we propose NMPC, a novel information-theoretic approach that learns a non-Markovian world model for planning in the latent space. We employ a recurrent contrastive predictive coding objective that specifically estimates the mutual information across temporal states. We show theoretically that our objective does not encourage the encoding of provably task-irrelevant information. This is critically different from reconstruction-based objectives as well as contrastive learning objectives that only measure the mutual information between the current observation and its latent code, which indiscriminately favor the encoding of both task-relevant and irrelevant information. Our experiments show that NMPC outperforms the state-of-the-art model when controls in environments dominated by task-irrelevant information while remaining competitive on standard control tasks from DMControl.

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

# A  HYPER PARAMETERS

## A.1  STANDARD SETTING

NMPC and Dreamer share the following hyperparameters:

**Model components**

- Latent state dimension: 30
- Recurrent state dimension: 200
- Activation function: ELU
- The action model outputs a tanh mean scaled by a factor of 5 and a softplus standard deviation for the Normal distribution that is then transformed using tanh (Haarnoja et al., 2018)

**Learning updates**

- Batch size: 50 for Dreamer and 250 for NMPC
- Trajectories length: 50
- Optimizer: Adam (Kingma & Ba, 2014) with learning rates $6 \times 10^{-4}$ for world model, $8 \times 10^{-5}$ for value and action model.
- Gradient update rate: 100 gradient updates every 1000 environment steps.
- Gradient clipping norm: 100
- Imagination horizon: 15
- $\gamma = 0.99$ and $\lambda = 0.95$ for value estimation

**Environment interaction**

- The dataset is initialized with $S = 5$ episodes collected using random actions.
- We iterate between 100 training steps and collecting 1 episode by executing the predicted mode action with $\mathcal{N}(0, 0.3)$ exploration noise.
- Action repeat: 2
- Environment steps: $2 \times 10^6$

Additionally,

- Dreamer clips the KL below 3 nats
- NMPC has a fixed set of coefficient in the overall objective for all control tasks: $\lambda_1 = 1, \lambda_2 = 0.1, \lambda_3 = 1, \lambda_4 = 1$. We use a fixed Gaussian noise $\epsilon \sim \mathcal{N}(0, 0.2^2)$ to add to the future latent code when computing CPC, as suggested in (Shu et al., 2020), and also use 0.2 as the fixed variance in instantaneous CPC.

In NMPC, we also use a target network for the value model and update this network every 100 gradient steps. Note that we also tried to use target value network for Dreamer, but it does not improve the results, as suggested by their original paper (Hafner et al., 2020).

## A.2  NATURAL BACKGROUND SETTING

To further encourage the model to focus on task-relevant information from observations, we increase the weight $\lambda_4$ of the reward loss in the training objective for both Dreamer and NMPC. In each control task they share the same reward coefficient, which is specified in the table below. All of the remaining hyperparameters are kept the same.

Table 1: Reward coefficients for different tasks in the natural backgrounds setting

| Task | Reward coefficient |
|------|--------------------|
| Cartpole Swingup | 1000 |
| Cheetah Run | 100 |
| Walker Run | 100 |
| Acrobot Swingup | 1000 |
| Hopper Hop | 100 |
| Cup Catch | 1000 |

## B  PROOF OF LEMMA 1

Our goal is to show that, under the conditions in Lemma 1,

$$\eta(\pi^*) \geq \eta(\pi^*_{\text{aux}}) \tag{16}$$

for any choice of auxiliary encoder $E'$.

We start by denoting $s_t = E^*(o_t)$ and $s'_t = E'(o_t)$. Note that the performance of $\pi$ can be written as

$$\eta(\pi_{\text{aux}}) = \mathbb{E}_{(\pi,p)} r(o_{1:T}) \tag{17}$$

$$= \sum_{\tau_{\text{aux}}} r(o_{1:T}) \prod_t p(o_t \mid o_{<t}, a_{<t}) p(s_t, s'_t \mid o_t) \pi_{\text{aux}}(a_t \mid s_{\leq t}, s'_{\leq t}, a_{<t}), \tag{18}$$

where $\tau_{\text{aux}}$ denotes the full trajectory of $(o, s, s', a)_{1:T}$ and $r(o_t)$ evaluates the reward at $o_t$ (for simplicity, we shall assume $p(r_t \mid s_t)$ is deterministic. Since $D_{\text{KL}}(p(r_t \mid o_t) \parallel R^*(r_t \mid E^*(o_t))) = 0$, we can rewrite as

$$\eta(\pi_{\text{aux}}) = \sum_{\tau_{\text{aux}}} R^*(s_{1:T}) \prod_t p(o_t \mid o_{<t}, a_{<t}) p(s_t, s'_t \mid o_t) \pi_{\text{aux}}(a_t \mid s_{\leq t}, s'_{\leq t}, a_{<t}), \tag{19}$$

where, with a slight abuse of notation, we note that $R^*(E^*(o_t)) = r(o_t)$. We now further rewrite $\pi_{\text{aux}}(a_t \mid s_{\leq t}, s'_{\leq t}, a_{<t})$ as

$$p(a_t \mid s_{\leq t}, s_{\leq t}, a_{<t}, \pi_{\text{aux}}), \tag{20}$$

and subsequently collapse the expression of the performance as

$$\eta(\pi_{\text{aux}}) = \sum_{(o,s,s',a)_{1:T}} R^*(s_{1:T}) p(o_{1:T}, s_{1:T}, s'_{1:T}, a_{1:T} \mid \pi_{\text{aux}}) \tag{21}$$

$$= \sum_{(s,s',a)_{1:T}} R^*(s_{1:T}) p(s_{1:T}, s'_{1:T}, a_{1:T} \mid \pi_{\text{aux}}), \tag{22}$$

where the last step arises from marginalization of $o_{1:T}$. Note by chain rule that $p(s_{1:T}, s'_{1:T}, a_{1:T} \mid \pi_{\text{aux}})$ becomes

$$\prod_t p(s_t \mid s_{<t}, s'_{<t}, a_{<t}, \pi_{\text{aux}}) p(s'_t \mid s_{\leq t}, s'_{<t}, a_{<t}, \pi_{\text{aux}}) p(a_t \mid s_{\leq t}, s'_{\leq t}, a_{<t}, \pi_{\text{aux}}). \tag{23}$$

By analyzing the Markov blankets in $p(s_{1:T}, s'_{1:T}, a_{1:T} \mid \pi_{\text{aux}})$, we can simplify the above expression to

$$\prod_t p(s_t \mid s_{<t}, s'_{<t}, a_{<t}) p(s'_t \mid s_{\leq t}, s'_{<t}, a_{<t}) p(a_t \mid s_{\leq t}, s'_{\leq t}, a_{<t}, \pi_{\text{aux}}). \tag{24}$$

Note that we omit the dependency on $\pi_{\text{aux}}$ in the first two terms since, given only the history of past actions and observations, the next observation does not depend on our choice of policy but only on the environment dynamics.

Since $E^*$ is optimal under the MI objective, we note that

$$I(S_{<t}, S'_{<t}, A_{<t}\,; S_t, S'_t) = I(S_{<t}, A_{<t}\,; S_t). \tag{25}$$

Eq. (25) implies that $s'_{<t}$ is independent of $s_t$ given $(s_{<t}, a_{<t})$, and that $(s_{<t}, s'_{<t}, a_{<t})$ is independent of $s'_t$ given $s_t$. This allow us to further simplify Eq. (24) to

$$\prod_t p(s_t \mid s_{<t}, a_{<t}) p(s'_t \mid s_t) \pi_{\text{aux}}(a_t \mid s_{\leq t}, s'_{\leq t}, a_{<t}). \tag{26}$$

Thus, the performance expression equates to

$$\eta(\pi_{\text{aux}}) = \sum_{\tau_{\text{aux}}} R^*(s_{1:T}) \prod_t p(s_t \mid s_{<t}, a_{<t}) p(s'_t \mid s_t) \pi_{\text{aux}}(a_t \mid s_{\leq t}, s'_{\leq t}, a_{<t}). \tag{27}$$

Note by way of similar reasoning (up to and including Eq. (24)) that

$$\eta(\pi) = \sum_{\tau} R^*(s_{1:T}) \prod_t p(s_t \mid s_{<t}, a_{<t}) \pi(a_t \mid s_{\leq t}, a_{<t}). \tag{28}$$

By comparing Eq. (27) and Eq. (28), we see that $s'_{1:T}$ effectively serves as a source of noise that makes $\pi_{\text{aux}}$ behave like a stochastic policy depending on the seed choice for $s'_{1:T}$. To take advantage of this, we introduce a reparameterization of $s'$ as $\epsilon$ such that

$$\eta(\pi_{\text{aux}}) = \sum_{\tau_{\text{aux}}} r(s_{1:T}) \prod_t p(s_t \mid s_{<t}, a_{<t}) p(\epsilon_t) \pi_{\text{aux}}(a_t \mid s_{\leq t}, \epsilon_{\leq t}, a_{<t}) \tag{29}$$

$$= \mathbb{E}_{p(\epsilon_{1:T})} \sum_{(s,a)_{1:T}} r(s_{1:T}) \prod_t p(s_t \mid s_{<t}, a_{<t}) \pi_{\text{aux}}(a_t \mid s_{\leq t}, \epsilon_{\leq t}, a_{<t}) \tag{30}$$

$$\leq \max_{\epsilon_{1:T}} \sum_{(s,a)_{1:T}} r(s_{1:T}) \prod_t p(s_t \mid s_{<t}, a_{<t}) \pi_{\text{aux}}(a_t \mid s_{\leq t}, \epsilon_{\leq t}, a_{<t}) \tag{31}$$

$$\leq \max_{\pi} \eta(\pi), \tag{32}$$

where the last inequality comes from defining a policy

$$\pi' := \pi_{\text{aux}}(a_t \mid o_{\leq t}, \epsilon^*_{\leq t}, a_{<t}) \tag{33}$$

and noting that the performance of $\pi^*_{\text{aux}}$ must be bounded by the performance of $\pi^*$. $\qquad \square$

# C ADDITIONAL RESULTS

In Figure 6, we show the comparison of NMPC with more baselines in the standard setting. They include Contrastive-Dreamer – the contrastive version of Dreamer reported in Hafner et al. (2020), and Markovian Predictive Coding (MPC) – in which we stack multiple frames and use a feed-forward dynamics instead of a recurrent one. The figure shows the results with two frames stacked.[1] MPC does not work in all of the control tasks. During training we observed that the CPC and Consistency objectives of MPC were much worse than those of NMPC. Therefore, we hypothesize that the feed-forward dynamics with stacked frames is not capable of predicting the far future, which leads to poor performance in tasks that require long-term planning.

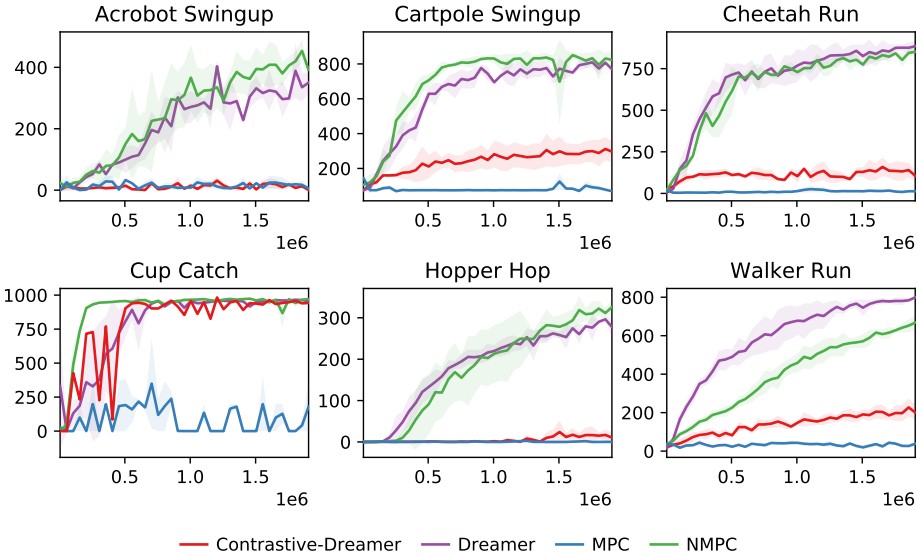

Figure 6: NMPC vs the baselines in the standard DMControl tasks. The return is computed in 1000environment steps. Each task is run with 3 seeds.

Figure 7 demonstrates the ablation analysis which compares NMPC with other contrastive variants, including Contrastive-Dreamer. It is easy to notice that all other models perform significantly worse than NMPC, verifying the necessary of different components that we proposed.

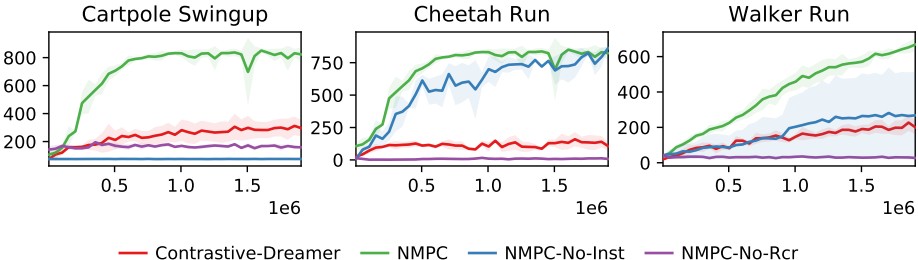

Figure 7: Abaltion study with more contrastive variants in the standard setting

---

[1]We also tried stacking three or four frames and observed similar performances.

In Figure 8 and 9, we run NMPC on more control tasks to make our results more comparable to Deep Bisimulation for Control (DBC) Zhang et al. (2020). We note that the gradient update rates of NMPC and DBC are different. In NMPC, we update 100 gradient steps after collecting 1000 environment steps, while DBC performs one gradient update after each environment step collected. Even with setup differences that put NMPC at disadvantage, our method either is competitive or vastly outperforms DBC.

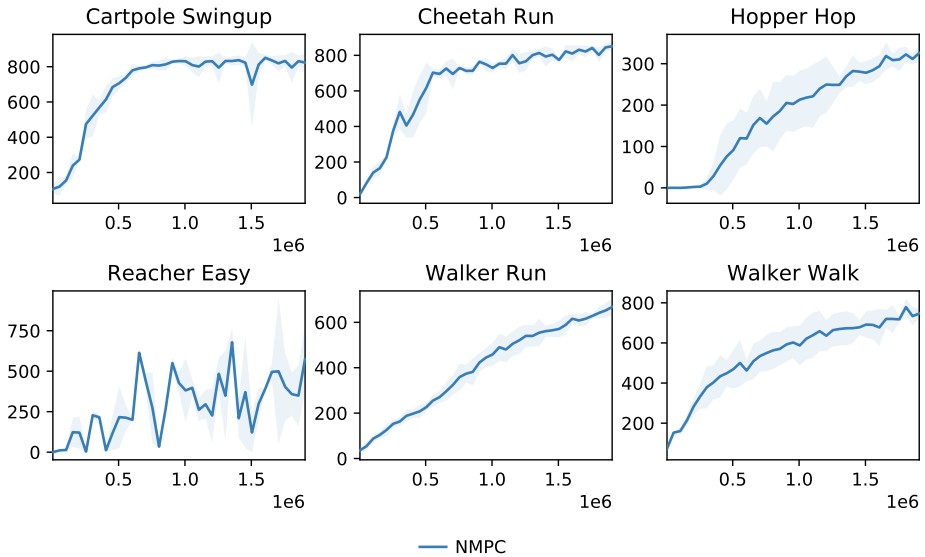

Figure 8: NMPC on more control tasks in the standard setting

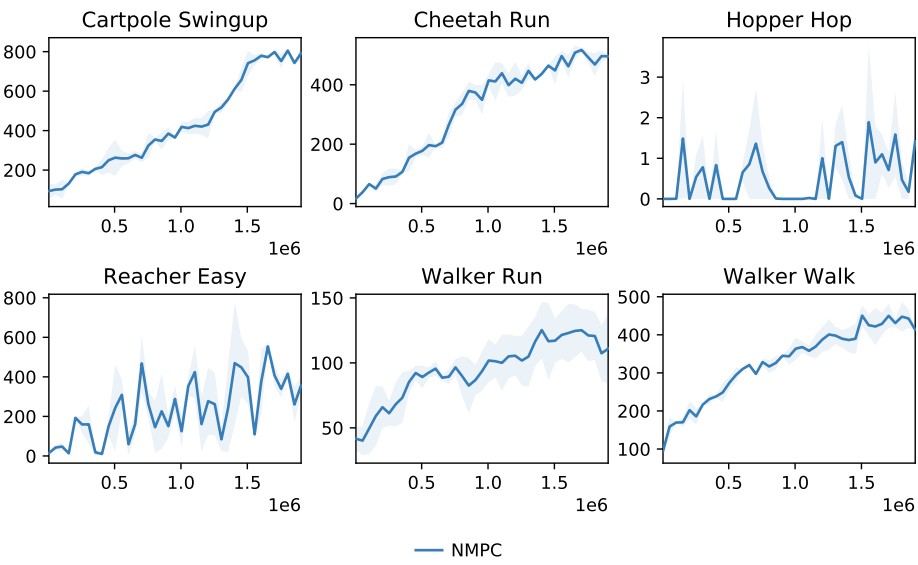

Figure 9: NMPC on more control tasks in the natural background setting

