# OpenReview forum: "Non-Markovian Predictive Coding For Planning In Latent Space"
_ICLR.cc/2021/Conference — Reject_

### Official Review · AnonReviewer2 · 2020-10-18
**Review 2: promising method, evaluation is lacking**

**Rating:** 5
**Confidence:** 5

**Review:**

---- Summary ----

The paper proposes a method for visual model-based reinforcement learning that relies on contrastive learning to learn the predictive model. Building on Hafner’20, the paper replaces the image reconstruction objective with a noise contrastive estimation (NCE) objective for the latent dynamics model, an NCE objective between the images and representations, and an additional maximum likelihood objective for the latent dynamics. It is shown that the method is competitive to Hafner’20 on the DM Control benchmark, and outperforms Hafner’20 on DM Control tasks with natural images used as background. The paper also theoretically analyzes one of the used objectives, arguing that it may lead to discarding irrelevant information.

---- Decision ----

The paper tackles a relevant and important problem of building predictive models that do not rely on image reconstruction, and proposes a promising method to this end. However, there are several technical weaknesses of the presented approach, and further important baselines and ablations are omitted. Due to lacking experimental evaluation, I lean toward rejecting the paper, but would be happy to update my score if the experimental evaluation were improved.

---- Strengths ----

The paper investigates a relevant and promising direction. The experiments show the effectiveness of the proposed method in simple continuous control scenarios, further supported by some theoretical analysis.

---- Weaknesses ----

The major weakness of the paper is a lack of representative baselines. The paper cites several similar papers, including Ding’20, and Shu’20, which the model is called “an extension of”. How is the proposed method better than these prior methods? Further, the central claim of the paper is the importance of non-markovian predictive coding, also claimed to be the main technical novelty (although this was already used in the foundational paper by van den Oord). It is never evaluated whether the non-markovian part improves performance. The contrastive learning version of Dreamer should be added to the plots as well.

A harder to fix issue is that the proposed method is rather unprincipled. Three different competing objectives are used (e.g. maximum likelihood of the latent dynamics reduces I(z;z’) while the other objectives increase it), and the interactions between them are not discussed. The paper presents only intuitive justification for the three objectives. This issue also causes the need to tune balance terms between different losses. For instance, the method requires tuning of the weight on the reward prediction, and requires different weights for different environments (Table 1 in the appendix), while in Dreamer this weight is always set to 1 and no hyperparameters are environment-specific. The weight for the recurrent CPC objective $\lambda_1$ does not seem to be specified in the paper.

---- Additional comments ----

Eq 1 - none of the symbols in the equation are defined.

### ---- Update ----

The authors' response does not satisfactorily address my concerns. My main concern is that the paper does not properly evaluate alternative choices, even though a large literature on contrastive learning exists.  While in the revision one baseline was added to the appendix, the main experiment still only contains a comparison to Dreamer and ablations. Further, it appears that the proposed method fails completely when the non-markovian part is removed. This is rather concerning since learning markovian latent dynamics is important and also possible with other methods (e.g. PlaNet-RNN). As far as I can tell, the paper does not discuss this issue and does not explain why learning non-markovian dynamics is crucial.

Overall, the paper proposes an interesting method but fails to provide any insight into how the method compares with possible alternatives. I, therefore, maintain my borderline score.

---

> ### Author Response · Authors · 2020-11-24
> **Response**
>
> Dear Reviewer,
>
> Thank you for recognizing the relevance and importance of this line of work.
>
> **Regarding how we improve upon PC3**
>
> As stated in the paper, the goal of our work is to extend PC3 to RL tasks. We note that the PC3 framework makes several design choices that make PC3 inapplicable to RL tasks such as the Deepmind Control suite:
>
> 1. PC3 assumed access to uniformly-sampled off-policy data. In contrast, NMPC takes the core representation learning machinery in PC3 (that of predictive coding and consistency) and showed that, by pairing with an actor-critic controller, it remains viable in an RL setting where the agent must continuously interact with, explore, and collect data from the environment.
> 2. PC3’s latent cost function is defined simply as the distance-to-goal-state in the latent space. Such a cost function made PC3’s latent space amenable to the application of locally-linear controller (which was the controller of interest in the PC3 paper). However, this also means it is difficult to apply PC3 to most DMC-level tasks where the reward function cannot be easily represented by a “goal state”. In contrast, NMPC explicitly ensures that the latent space is predictive of the reward function so that the latent space is amenable to other types of controllers, i.e. actor-critic.
>
> **Regarding use of Markovian vs Non-Markovian dynamics model**
>
> In addition to these inherent limitations of PC3 stated above, we also wish to address the importance of using non-Markovian dynamics model in NMPC. At the reviewer’s request, we have included experiments comparing the performance of our model when using a Markovian vs non-Markovian dynamics model. In Figure 6 in Appendix C we show that replacing the non-Markovian dynamics model with a Markovian dynamics model leads to significantly worse performance.
>
> **Regarding our relation to Ding ‘20**
>
> Although Ding ‘20 concurrently developed a similar model, we believe our work distinguishes itself from theirs on several front:
>
> 1. In terms of theory, our paper uniquely provides a careful theoretical characterization of how exactly temporal predictive coding relates to the encoding of task relevant vs irrelevant information. Our Lemma 1 and the ensuing remarks we included Section 3.2 were specifically intended to “set the record straight” on what are the exact benefits and limitations of using temporal predictive coding. Please see our response to Reviewer 3 for a more in-depth discussion on the task-relevance bias of temporal predictive coding.
> 2. In terms of experiments, the distractors used in Ding ‘20 (basic shapes) were far less complicated than the natural video distractors considered in our work.
> 3. There are also a number of other more subtle differences between our work, including: A) It is unclear how they collected samples in order to estimate an action-conditional mutual information objective. In contrast, our work lower bounds the mutual information objective $I(E(O_t) ; E(O_{<t}), A_{<t})$ jointly over $(E(O_{<t}), A_{<t})$ and shows how the negative samples can be efficiently constructed for our estimator. B) Whereas Ding ‘20 makes use of an auxiliary bilinear critic, our work draws inspiration from the PC3 framework and directly uses the learned dynamics model as the critic.
>
> **Regarding our relation to contrastive Dreamer**
>
> The key difference between NMPC and contrastive Dreamer is that NMPC estimates the mutual information across temporal latent states (which we call “temporal predictive coding”), whereas contrastive Dreamer estimates the mutual information only between the current observation and its latent representation (which we call *instantaneous predictive coding*). This is a critical difference, since our theoretical analysis only holds for temporal predictive coding, and not instantaneous predictive coding.
>
> At the reviewer’s request, we have also included a comparison to contrastive Dreamer in Figures 6, which shows that NMPC out-performs contrastive Dreamer significantly.

---

> > ### Author Response · Authors · 2020-11-24
> > **Response (cont'd)**
> >
> > **Regarding the “competing” objectives and hyperparameter tuning**
> >
> > The phenomenon of the consistency objective causing map shrinkage is extensively detailed in Shu ‘20. While the consistency and information-theoretic objectives “compete” to counter the stability of the latent map size, Shu ‘20 found, however, that the dynamics model learned purely via consistency maximization nevertheless serves as a good critic for the CPC objective (see Table 2 of Shu ‘20). This observation is consistent with our understanding that the optimal dynamics model is an optimal critic. As such, once a suitable set of hyperparameters is chosen to ensure map stability, no further action needs to be taken.
> >
> > Importantly, we used the same choice of $(\lambda_1, \lambda_2, \lambda_3)$in *all* experiments (both in the standard and natural video settings), indicating that our hyperparameters for the consistency vs information-theoretic objectives are highly robust to environment changes. We apologize for accidentally omitting this information in our initial submission and have revised our submission accordingly.
> >
> > Furthermore, we wish to emphasize that we did not tune lambda4 in the standard setting (it was fixed to $\lambda_4 = 1$). In the natural background setting, however, we tuned the reward coefficient for all models (including Dreamer) to ensure a fair comparison.
> >
> > Thus, from a hyperparameter tuning perspective, we believe NMPC incurs no greater difficulty than Dreamer beyond the initial tuning of $(\lambda_1, \lambda_2, \lambda_3)$ to ensure map stability.
> >
> > **Regarding additional comment**
> >
> > Thank you for pointing this out. We will correct it.

---

### Official Review · AnonReviewer4 · 2020-10-26
**Simple and clean framework; Novelty concern**

**Rating:** 6
**Confidence:** 4

**Review:**

Summary:

This paper proposes an information-theoretic framework for learning a world model that encodes task-relevant information of the world. It shows that the learned encoder and dynamics model can be used to train the policy and fitting the value function to agent to perform comparibly well to Dreamer on standard tasks and outperform them when there are distractions in the scene. The paper also provides a theoretical anylisis of the task-relevant information in the encoding.

Pros:

This paper provides a clean framework that learns the embeddings that contain task-relevant information.

I appreciate the theoretical formulation to show that the optimal embedding will containis sufficient to train the policy on. I believe this theoretical analysis is novel.

The experiments provide an insightful ablation and show that the proposed contrastive model is less agnostic to the background distraction.

Cons:

My main conern is regarding the novelty. This paper misses a few recent relevant work [1, 2, 3]. These work also use contrastive losses to learn a latent world model and outperforms Dreamer on DMControl tasks and natural background. I'd like to see the comparison against [1,2] or clarify how NMPC is different and/or better in some ways.
How do you think NMPC performs against this model-free version with similar task-relevant objective [4]?

In the orignal paper dreamer also proposes a contrastive loss for training the latent model although it is not temporal. Is the ablation NMPC-No-Rcr in figure 2 comparable to that or different? Also, how important is having a recurrent dynamics model as supposed to a feedforward dynamics model?

How senstive the algorithm is to the hyperparameter combinations? The loss seems to have a few terms to be tuned. It would be helpful if the ablation can show the importance of other components such as Non-Markovian consistency and dynamics smoothing.

Conclusion:

Overall, the paper is well written and easy to understand. It proposes a simple framework with experimental and theoretical support. The main downsight lies in its novelty as a few other works that aim to tackle this issue in a similar way. If this can be addressed, I think it is a really good paper.

References:

[1] Dreaming: Model-based Reinforcement Learning by Latent Imagination without Reconstruction (https://arxiv.org/abs/2007.14535)

[2] Contrastive Variational Model-Based Reinforcement Learning for Complex Observations (https://arxiv.org/abs/2008.02430)

[3] Learning Predictive Representations for Deformable Objects Using Contrastive Estimation (https://arxiv.org/abs/2003.05436)

[4] Learning Invariant Representations for Reinforcement Learning without Reconstruction (https://openreview.net/pdf?id=-2FCwDKRREu_

---

> ### Author Response · Authors · 2020-11-24
> **Response**
>
> Dear Reviewer,
>
> Thank you for recognizing the novelty of our theoretical formulation and cleanliness of the framework. We hope to address your concerns regarding the novelty of our proposed model. Thank you also for bringing our attention to several related works; we will include them in our revision of the paper. As the reviewer mentioned, [1, 2] are closely related to our work and it is worth explaining how our work differs.
>
> **Regarding relation to [1]**
>
> In comparison to [1], there are several theoretical and experimental design differences. Most importantly, our work is motivated by temporal predictive coding’s unique ability to dismiss unpredictive features in comparison to reconstruction-based loss; our paper provides both theoretical and experimental results specifically regarding this hypothesis. In contrast, [1] motivates their work as a means of overcoming an “object vanishing” phenomenon that they observed in the original reconstruction-based Dreamer. To our understanding, this is an orthogonal consideration and thus resulted in a different set of experimental designs in their paper.
>
> From an implementation standpoint, we also differ from [1] in two notable ways: 1) whereas [1] needed to introduce an auxiliary “linear forward dynamics” model as part of the critic for the CPC objective, we instead showed that it is viable to simply reuse the learned dynamics model itself as the critic. 2) a major component of [1] is the use of data augmentation. In contrast, since our goal is to scientifically verify the value of temporal predictive coding in contrast to reconstruction-based approaches, we did not use data augmentation since it would introduce a confounder. Indeed, our experiments show that data augmentation is unnecessary for NMPC to be competitive with Dreamer in the standard setting and superior to Dreamer in the natural background setting.
>
> **Regarding relation to [2]**
>
> In comparison to [2], the key difference is in the information-theoretic object that we seek to optimize. Our work is specifically interested in the mutual information across temporal states, $I(E(O_t) ; E(O_{<t}), A_{<t})$, and thus resulting in a temporal predictive coding objective. In contrast, [2] only considers the mutual information $I(O_t; S_t)$ between the current observation and its corresponding latent representation. This mutual information quantity considered in [2] is problematic since it does not benefit from our theoretical analysis, which formally characterizes *temporal* predictive coding’s ability to dismiss unpredictive (and thus task-irrelevant) features.
>
> Oddly, the mutual information quantity $I(O_t; S_t)$ has been reported to perform poorly experimentally both in the original Dreamer paper and in our paper (Figure 5, 6, and 7. The latter two figures were added to the appendix as part of this rebuttal). It remains unclear why [2] does not exhibit a similar poor behavior; given the numerous modifications introduced in [2], it is unfortunately difficult to pinpoint the source of this experimental discrepancy.
>
> Overall, our work proposes a well-motivated temporal objective, carefully characterizes the theoretical benefits and limitations of temporal predictive coding for removing task-irrelevant features, and offers a clean suite of experiments specifically to test this hypothesis.
>
> **Comparison to task-relevance objective in [4]**
>
> Thank you for bringing our attention to [4]. In terms of performance, we have expanded our experiments to include additional DMC tasks (see Figures 8 and 9 in Appendix C) in order to assist the comparison with [4]. By comparing our Figures 8/9 with their Figures 11/13, we can see that NMPC exhibits competitive (and often superior) performance on both the standard and natural video tasks.
>
> Interestingly, we wish to note that the mechanism by which our objectives promote robustness to task-irrelevant features is quite different. Whereas [4] specifically targets the reward function to filter task-irrelevant features, our temporal predictive coding objective only filters unpredictive features. The empirical success of temporal predictive coding in comparison to [4] *despite* not having access to the reward structure shows the efficacy of filtering out unpredictive/hard-to-predict features.
>
> It is also possible that the filtering of unpredictive features is an inherently easier objective than approximating the bisimulation metric as done in [4]. If this is the case, then we think NMPC will serve as a strong baseline and perhaps can be used in conjunction with task-aware objectives to achieve even better results in the future.

---

> > ### Author Response · Authors · 2020-11-24
> > **Response (cont'd)**
> >
> >
> > **Regarding contrastive dreamer and additional ablations**
> >
> > At the reviewer’s request, we have included Contrastive Dreamer in Figures 6 and 7. We wish to note that both Contrastive Dreamer and NMPC-No-Rcr use a contrastive predictive coding objective that lower bounds the mutual information quantity $I(O_t; S_t)$, though there are some slight differences in terms of how we define the generation process of $(O_t, S_t)$; whereas Contrastive Dreamer used a recurrent dynamics model to define this generation process, we simply used a feedforward encoder that directly maps $O_t$ to $S_t$. These differences may be why Contrastive Dreamer achieves better performance than NMPC-No-Rcr in Figure 7, though both models ultimately perform quite poorly in comparison to NMPC.
> >
> > We have also included experiments comparing the performance of our model when using a Markovian vs non-Markovian dynamics model. In Figure 6 in Appendix C we show that replacing the non-Markovian dynamics model with a Markovian dynamics model leads to significantly worse performance, indicating that the non-Markovian dynamics model plays a critical role in NMPC’s performance.
> >
> > **Regarding robustness of hyperparameters**
> >
> > We used the same choice of $(\lambda_1, \lambda_2, \lambda_3)$ in *all* experiments (both in the standard and natural video settings), indicating that our hyperparameters for the consistency vs information-theoretic objectives are highly robust to environment changes.
> >
> > Furthermore, we wish to emphasize that we did not tune lambda4 in the standard setting (it was fixed to $\lambda_4 = 1$). In the natural background setting, however, we tuned the reward coefficient for all models (including Dreamer) to ensure a fair comparison.
> >
> > Thus, from a hyperparameter tuning perspective, we believe NMPC incurs no greater difficulty than Dreamer beyond the initial tuning of $(\lambda_1, \lambda_2, \lambda_3)$ to ensure map stability (for a more in-depth discussion on how $(\lambda_1, \lambda_2, \lambda_3)$ is related to map stability, please see our response to Revewier 2 on *Regarding the “competing” objectives and hyperparameter tuning*).

---

### Official Review · AnonReviewer3 · 2020-10-27
**Interesting work using information theory for model-based RL**

**Rating:** 6
**Confidence:** 3

**Review:**

Summary:
The paper introduces a new method for model based RL that learns a dynamical latent representation from pixel data (images) using a maximum mutual information criterion together with a predictability loss. The core idea is that maximizing mutual information between states bias the encoder toward learning predictable (and therefore potentially task relevant) latent features while discarding unpredictable features.

Relevance:
The paper addresses the very relevant problem of learning representations usable in a control task.

Originality:
The core novelty of the paper is to combine the mutual information approach introduced with PC3 in the context of control theory with the "dream to control" approach for model-based reinforcement learning in pixel space.

Scientific quality:
- The proposed approach is in general well motivated. However. I am not convinced by the emphasis that the authors put in the proposition that the maximum mutual information loss helps to learn task relevant features. While it is true that the proposed approach is biased towards temporally predictable features, most distinctive features both in the real wold and in most games and simulations have as much temporal predictability as the task relevant ones. For example, the video backgrounds in the experiments are completely task irrelevant and at the same time highly predictable. In general, the encoder cannot trulely promote task-relevant features without having access to the reward structure.
- The experiment section offers a decently wide range of experiments. However, the authors should include more baselines, possibly including other model-based methods such as [1] and model-free methods such as some variant of DQNs.

Pros:
- Very relevant research area
-Rather original combination of methods
-Clear and well-written paper

Cons:
- The main claim that the method is biased towards learning task-relevant features is questionable.
-  The experiments should contain more baselines including other model based approaches and some model free approach.

References:
[1] Hafner, Danijar, et al. "Learning latent dynamics for planning from pixels." International Conference on Machine Learning. PMLR, 2019.

---

> ### Author Response · Authors · 2020-11-24
> **Response**
>
> Dear Reviewer,
>
> Thank you for recognizing the relevance and novelty of our contributions.
>
> **Regarding the limitations of the task-relevance bias of predictive coding**
>
> Thank you for raising an important point regarding the theoretical limits of what predictive coding can do in terms of biasing the representations toward task-relevant features. Our theoretical analysis formally shows that:
>
> 1. Temporal predictive coding does not encourage the encoding of unpredictive features (this is in contrast to reconstruction-based approaches).
> 2. Unpredictive features are task-irrelevant.
>
> However, as the reviewer observed: while all unpredictive features are task-irrelevant, not all task-irrelevant features are unpredictive. This is an important distinction and one which we explicitly made note of in our paper as well (see the final paragraph in Section 3.2).
>
> Even with this caveat in mind, we wish to make several arguments in favor of our work:
> 1. We show, from a theoretical perspective, that (temporal) predictive coding is still strictly superior to reconstruction-based approaches---since predictive coding will dismiss unpredictive (and thus task-irrelevant) features whereas reconstruction-based approaches will not.
>
> 2. We show, empirically, that NMPC outperforms Dreamer on natural video background tasks. We believe this empirical result is of scientific value since it suggests one of two possibilities: 1) either natural videos do in fact contain considerable amounts of unpredictable information that NMPC successfully dismisses, or 2) natural videos contain considerable amounts of hard-to-predict information that our dynamics model deems effectively-unpredictable. The latter possibility is intimately related to the concept of “usable information” (Xu, et al., 2020) whenever we instantiate a mutual information estimator with finite-computational power.
>
>  We agree that it is important to be mindful of these two distinct possibilities, especially in bridging the gap between our theoretical analysis and our empirical design. We will include a more thorough discussion of these considerations in our paper. While not shown in our paper, our preliminary experiments were in fact conducted using natural backgrounds where each frame was drawn i.i.d. (and thus completely unpredictable). We are happy to include these results as well to give the reader a broader context of how the empirical results relate to our theoretical claims.
>
> 3. As the reviewer noted, there are inherent limits to what an encoder can filter if the representation objective does not have access to the reward structure. However, the empirical success of temporal predictive coding *despite* not having access to the reward structure shows the efficacy of filtering out unpredictive/hard-to-predict features. Even as we move forward in developing task-aware information-theoretic objectives, we think NMPC will serve as a strong baseline and perhaps can be used in conjunction with task-aware objectives to achieve even better results in the future.
>
> Reference: Yilun Xu, Shengjia Zhao, Jiaming Song, Russell Stewart, Stefano Ermon. A Theory of Usable Information Under Computational Constraints. ICLR 2020.
>
> **Regarding the inclusion of more baselines**
>
> Since our primary goal was to extend the PC3 framework to be able to handle DMC-level tasks, we decided to compare against the current best model-based RL method, Dreamer. We did not compare against PlaNet as it was reported to be inferior to Dreamer on most of the control tasks in the Dreamer paper.
>
> As part of our rebuttal, we have expanded our experiments to include additional DMC tasks (see Figures 8 and 9 in Appendix C) in order to assist readers who wish to compare NMPC to other works.
>
> For example, in comparison to a concurrent submission (https://openreview.net/forum?id=-2FCwDKRREu) on learning task-relevant features paired with a model-free controller, comparison of our Figures 8/9 with their Figures 11/13 shows that NMPC exhibits competitive (and often superior) performance on both the standard and natural background tasks.

---

> > ### Author Response · Authors · 2020-11-24
> > **Response (cont'd)**
> >
> > **Regarding additional scientific merits**
> >
> > We also wish to highlight to the reviewer that the scientific contributions in the paper go beyond the testing of NMPC in tasks that include distractors. Since our goal is to scale PC3 to Dreamer-level tasks, several other conclusions can be drawn from our work:
> > 1. PC3 was specially designed for locally-linear controllers. By replacing the LQR controller with the Actor Critic controller used in Dreamer, we demonstrated that (predictive coding and consistency) suffices as the core representation learning machinery in model-based reinforcement learning when using a contrastive learning method.
> > 2. PC3 assumed access to uniformly-sampled off-policy data. We showed that the core representation learning machinery in PC3 (that of predictive coding and consistency) remains viable in an RL setting where the agent must continuously interact with, explore, and collect data from the environment.
> > 3. PC3 used a Markovian dynamics model. As part of our rebuttal, we also addressed the significance of the non-Markovian dynamics model in our NMPC model. In Figure 6 in Appendix C shows that replacing the non-Markovian dynamics model with a Markovian dynamics model leads to significantly worse performance.

---

### Official Review · AnonReviewer1 · 2020-10-28
**PREDICTIVE CODING FOR PLANNING IN LATENT SPACE**

**Rating:** 5
**Confidence:** 5

**Review:**

Problem Setup: The paper proposes a mutual information objective to learn a latent representation which
can be used for planning. The paper note that most of the existing model based RL methods learn a
model of the world via reconstruction objective, which requires to predict each and every detail of the visual
input, and hence can be detrimental in case of noisy inputs or in the presence of distractors.

Proposed idea: In order to tackle this problem, the paper proposes a mutual information objective to maximize
the mutual information between the latent codes at distinct time steps.  In order to capture the history of the past,
the authors utilize a recurrent model (from Dreamer Model) to encode information about the history of the trajectory.
The paper also uses 2 different objectives in order to prevent the representation from collapsing. The paper proposes
to use a mutual information objective between the observation and the encoding of the observation (as in Dreamer),
as well as consistency objective in the latent space (already used before). Essentially the underlying idea behind the proposed method is not new per se, but as far as I know this is the first paper, which has shown to make it work on DeepMind Control tasks.

Experiments: The authors compare the proposed method to DREAMER model on 6 DeepMind Control (DMC) tasks.
The authors also evaluate the robustness of the proposed method by evaluating the capability of the proposed method
in dealing with complicated backgrounds (given the scenario, when the entity of interest occupies a small region in the input).

Clarity: The paper is clearly written.

References: Their are bunch of references that could be cited. Shaping belief paper [1] also uses a CPC style objective for learning a model of the environment. [2] also learns a model of the environment by predicting only the relevant information by constructing a temporal information bottleneck but still within the framework of maximum likelihood prediction. [3] also uses a mutual information based objective and without any explicit reconstruction.

- [1] Shaping Belief States with Generative Environment Models for RL https://arxiv.org/abs/1906.09237
- [2] Learning dynamics model in reinforcement learning by incorporating the long term future https://arxiv.org/abs/1903.01599
- [3] Dreaming: Model-based Reinforcement Learning by Latent Imagination without Reconstruction
 https://arxiv.org/abs/2007.14535

Scalability: It would be interesting to see how the proposed method evaluates on more challenging tasks just as on atari or on continuous control tasks such as "box" stacking which requires some relational reasoning. Since the underlying idea has been tried in some other context and in this work the contribution is to make it work for deep RL problems, it becomes important to evaluate on more challenging problems and tasks.

======

After Rebuttal: I have read the rebuttal, as well as reviews by other reviewers. I keep my original score. Hope to see a better version of the paper soon.

---

> ### Author Response · Authors · 2020-11-24
> **Response**
>
> Dear Reviewer,
>
> Thank you for recognizing our work as the first to scale the combination of information-theoretic and latent space prediction approaches to the DeepMind control suite tasks.
>
> **Regarding the contributions in our work**
>
> The core motivation of our work was to take the PC3’s (predictive coding, consistency, and curvature) stochastic optimal control framework and assess how to adapt the framework to DMC-level tasks. In doing so, our paper addresses several important scientific questions:
>
> 1. PC3 was specially designed for locally-linear controllers. By replacing the LQR controller with the Actor Critic controller used in Dreamer, we demonstrated that (predictive coding and consistency) suffices as the core representation learning machinery in model-based reinforcement learning when using a contrastive learning method.
> 2. PC3 assumed access to uniformly-sampled off-policy data. We showed that the core representation learning machinery in PC3 (that of predictive coding and consistency) remains viable in an RL setting where the agent must continuously interact with, explore, and collect data from the environment.
> 3. Robustness of predictive coding to distractors: by scaling to DMC-level tasks and exploring natural video backgrounds, we were able to empirically demonstrate the robustness of predictive coding to distractors, a phenomenon that went underappreciated in the original PC3 paper. Furthermore, we uniquely provide a formal theoretical characterization of this phenomenon: showing that predictive coding does not encourage the encoding of unpredictive features and proving that unpredictive features are task-irrelevant information.
> 4. PC3 used a Markovian dynamics model. As part of our rebuttal, we also addressed the significance of the non-Markovian dynamics model in our NMPC model. In Figure 6 in Appendix C we show that replacing the non-Markovian dynamics model with a Markovian dynamics model leads to significantly worse performance.
>
> We fully agree that, by showing that the contrastive model-based RL can scale to DMC-level tasks, the next exciting direction is to further scale this approach to tackle Atari/continuous control tasks. With that in mind, we believe that our existing contributions are significant in paving the way for such subsequent works on contrastive model-based RL.
>
> **Regarding additional references**
>
> Thank you for bringing our attention to these additional papers. We will be sure to include them in our revised paper. For now, we wish to note several key differences that distinguish us from these additional works:
> 1. Paper [1] employs a temporal CPC objective to learn the latent space. However the proposed method is model-free, and the dynamics is discarded during control. This leads to the second distinction: our work uses the dynamics as the critic in CPC objective, whereas they use an additional classifier to distinguish positive and negative pairs
> 2. Paper [2] is similar to ours in terms of learning latent dynamics for control. However, they use a reconstruction based lower bound combined with an auxiliary loss to predict longer term future, while we focus on developing a decoder-free method.
> 3. Paper [3] also uses a temporal contrastive objective $I(O_t; S_{t-k})$ to learn the latent dynamics for control. However, they require training an independent linear dynamics model solely to roll out into the future and compute the contrastive loss, which is then discarded during control. Moreover, they use a bilinear model as the critic while we use the learned dynamics model itself. They also use data augmentation of the observations, while we do not.

---

### Author Response · Authors · 2020-11-24
**Rebuttal summary**

Dear reviewers,

Thank you for your feedback and insightful comments. We hope that our responses satisfactorily address your questions. Based on the reviews, we wish to highlight two key considerations that distinguish us from existing/concurrent works.

1. Our paper is specifically interested in and successfully tackles the challenge of scaling PC3 to DMC-level reinforcement learning tasks (please see our response to Reviewer 1 for an enumeration of the improvements we made to PC3 and the consequences of those improvements). As requested by Reviewer 2 and 4, we have also included ablations showing the importance of using non-Markovian dynamics in comparison to the Markovian dynamics model employed in PC3.

2. We believe our paper offers the first due-diligence characterization of how temporal predictive coding differs from reconstruction-based approaches. As noted by Reviewer 4, we presented the first formal analysis showing that 1) temporal predictive coding is capable of filtering out unpredictive information, and that 2) unpredictive information is task-irrelevant. Notably, this analysis does not apply to non-temporal (i.e. instantaneous) predictive coding, and we hope our analysis and experimental results will persuade practitioners to specifically consider the temporal variants of predictive coding instead when performing model-based RL.

 We are also, however, careful to acknowledge the limitations of temporal predictive coding. As noted in Section 3.2 of our paper and by Reviewer 3, since temporal predictive coding is task-agnostic, it will still encourage the encoding of predictive but task-irrelevant information. Nevertheless, the empirical success of NMPC (especially when compared to https://openreview.net/forum?id=-2FCwDKRREu, which filters information in a task-aware manner) indicates the surprising efficacy of filtering out unpredictive/hard-to-predict features. Even as we move forward in developing task-aware objectives, we think NMPC will serve as a strong baseline and perhaps can be used in conjunction with task-aware objectives to achieve even better results in the future.

  Please see our response to Reviewers 3 for an in-depth discussion on our characterization of temporal predictive coding and how we bridge the gap between our theoretical analysis and our experiments.

In addition to these two key considerations, several reviewers requested comparisons to specific related works and also inquired about the ease of hyperparameters. We hope that our response to each reviewer satisfactorily explains how we differ from the related works. As for hyperparameter tuning, we wish to reiterate that, with the exception of the reward coefficient in the natural background setting (which we tuned for both NMPC and Dreamer), all hyperparameters are kept the same in the experiments (please see our response to Reviewer 2 for a detailed discussion on this matter).

---

### Decision · Program_Chairs · 2021-01-07
**Final Decision**

**Decision:**

Reject

**Comment:**

This paper presents Non-Markovian Predictive Coding (NPMC), a method for learning state representations in visual RL domains that can be used for planning. This work builds on recent work on PC3 (Shu et al. 2020) and PlaNet (Hafner et al. 2020). Concretely, NPMC replaces the image reconstruction objective in PlaNet with a noise contrastive estimation (NCE) objective for the latent dynamics model, an NCE objective between the images and representations, and an additional maximum likelihood objective for the latent dynamics.

Reviewers were in agreement that this paper tackles an important problem and appreciated the writing quality, the experiments that demonstrate effectiveness of NPMC in continuous control scenarios, and accompanying theoretical analysis. However, reviewers were on balance in consensus that this paper needs another iteration before it can appear. Aside from discussion of related work, the main weakness noted by reviewers is that the manuscript in its current form makes it clear how NPMC differs from closely related methods from a technical point of view, but does not make it sufficiently clear to what extent non-Markovian predictive coding leads to improved planner performance. In particular, the paper lacks detailed comparisons to baselines, and reviewers were not sufficiently convinced by experiments that were added to the appendix after discussion.

The authors indicate that their contribution is that NPMC extends PC3 to RL tasks. The metareviewer appreciates that experimental comparisons can require creative thinking when baselines are not directly applicable to the tasks of interest, but would nonetheless like to encourage the authors to consider how they can improve their experiments.